# Training stochastic stabilized supralinear networks by dynamics-neutral growth

**Wayne W.M. Soo**
Department of Engineering
University of Cambridge
wmws2@cam.ac.uk

**Máté Lengyel**
Department of Engineering     Department of Cognitive Science
University of Cambridge       Central European University
m.lengyel@eng.cam.ac.uk

## Abstract

There continues to be a trade-off between the biological realism and performance of neural networks. Contemporary deep learning techniques allow neural networks to be trained to perform challenging computations at (near) human-level, but these networks typically violate key biological constraints. More detailed models of biological neural networks can incorporate many of these constraints but typically suffer from subpar performance and trainability. Here, we narrow this gap by developing an effective method for training a canonical model of cortical neural circuits, the stabilized supralinear network (SSN), that in previous work had to be constructed manually or trained with undue constraints. SSNs are particularly challenging to train for the same reasons that make them biologically realistic: they are characterized by strongly-connected excitatory cells and expansive firing rate non-linearities that together make them prone to dynamical instabilities unless stabilized by appropriately tuned recurrent inhibition. Our method avoids such instabilities by initializing a small network and gradually increasing network size via the dynamics-neutral addition of neurons during training. We first show how SSNs can be trained to perform typical machine learning tasks by training an SSN on MNIST classification. We then demonstrate the effectiveness of our method by training an SSN on the challenging task of performing amortized Markov chain Monte Carlo-based inference under a Gaussian scale mixture generative model of natural image patches with a rich and diverse set of basis functions – something that was not possible with previous methods. These results open the way to training realistic cortical-like neural networks on challenging tasks at scale.

## 1   Introduction

Biological neural network models of the brain have long been studied to understand some of the fundamental network mechanisms employed by the brain [1–4]. However, these models were not capable of actually achieving the brain's performance on its wide range of complex computations. Conversely, artificial neural networks have been achieving competitive performance in a multitude of image, language, and speech-related tasks, but typically without regard to biological realism [5–8]. In systems neuroscience, neural networks have also been trained on laboratory tasks employed in typical experiments [9–16]. These approaches provided important insights into the contributions of experimentally found macroscopic neural dynamics and representations to the successful performance of such tasks. However, these networks did not incorporate some of the most salient constraints on the detailed organization of cortical circuits. For example, they were purely feedforward [10], utilized neuronal transfer functions that were either outright saturating (e.g. tanh) or at best lacked the superlinear characteristics of cortical neurons (as in ReLUs, or the rectified softplus) [9–11, 15, 16], had noiseless dynamics [12], no separation of excitatory (E) and inhibitory (I) cells [9, 10, 15], and sometimes even employed artificial gating mechanisms such as LSTMs [13]. Due to these properties,

36th Conference on Neural Information Processing Systems (NeurIPS 2022).

these networks remained abstracted away from biological neurons in key aspects, and hence offered limited insight into the neuron-level mechanisms that drive their computations.

To bring the modeling of neural circuit mechanisms underlying challenging computations at (or near) single-cell resolution within reach, we develop a novel method to train an experimentally-supported neural network model, the stabilized supralinear network (SSN) [17]. Critically, the SSN satisfies many of the key constraints of cortical circuits: it has separate but recurrent excitatory and inhibitory populations, expansive (rectified power-law) single neuron non-linearities, realistic single-neuron time constants, and no reliance on artificial gating mechanisms. In its original form, the SSN continues a venerable tradition of handcrafted excitatory-inhibitory (E-I) networks whose dynamics have been extensively studied [18, 19], and that helped reveal key consequences of the cortex operating in a balanced E-I regime [3, 20–22]. In particular, the SSN accounts for the experimentally-observed effects of stimulus tuning, sublinear response summation and surround suppression of neural responses in sensory cortices [4, 23], as well as oscillations, bistable and persistent responses [24]. Furthermore, when extended to include noise, the resultant stochastic SSN produces realistic, stimulus-modulated population-wide patterns of noise variability [25].

However, the same features that make the SSN an attractive substrate for its biological realism also make it particularly challenging to train it. This is because, in SSNs, excitatory cells are typically strongly connected with one-another in order to implement non-trivial nonlinear transformations of their inputs, such as divisive normalization, which underlie many of the experimentally observed phenomena for which they account. In addition, single neuron non-linearities are expansive to reflect the experimental finding that cortical neurons almost exclusively use the convex (non-saturating) part of their firing rate non-linearities under physiological conditions, including for the strongest stimuli [26] (even if they can be trivially driven to saturation by direct current injection, due to the refractory period of the mechanism generating action potentials). These properties make SSNs particularly susceptible to dynamical instabilities resulting in run-away excitation, thus rendering their training highly challenging. Indeed, in the few cases in which the training of SSNs was attempted, either noiseless neurons were used [27, 28], or the network was so heavily under-parameterized that it substantially limited its expressivity [29].

Here, we develop a new method to train SSNs that avoids dynamical instabilities during training and is able to train SSNs at scale on a variety of tasks. While standard approaches to training networks only optimize weights and keep the architecture of the network fixed throughout network training (but see recent work in Refs. 30–32), our method alternates between optimizing network weights with a fixed architecture, and changing the architecture by growing the network. Importantly, in each network growth step, new neurons are added such that they do not affect the dynamics of the network before weights are optimized again. We first demonstrate the effectiveness of our method by training an SSN on a standard machine learning benchmark, MNIST classification [33]. We achieve only slightly lower accuracies than state-of-the-art, despite restricting our network to be biologically plausible. For a direct comparison, we also train an SSN on a probabilistic inference task that has been shown to have neurobiological relevance but has proved to be a challenging target [29]. Our approach successfully trains an SSN that has three orders of magnitude more parameters than what was previously possible to optimize. The trained SSN performs accurate inference under a Gaussian scale mixture-model (GSM) [34] with a set of basis functions that is sufficiently rich for allowing high-quality reconstructions of CIFAR-10 [35] images. Inference under this GSM represents an unachievable target for previous, heavily constrained approaches for training SSNs. These results validate our approach and open the way to training SSNs on a large variety of different tasks, and to using the resulting networks to study the circuit mechanisms of neural computations at a single cell resolution.

## 2    Methods

### 2.1    Stochastic supralinear stabilized network

The SSN is a canonical model of cortical circuits that has been shown to account for a wealth of neural response properties in the primary visual cortex [17, 4, 25]. Its dynamics are described by

$$\boldsymbol{\tau} \frac{\mathrm{d}\mathbf{u}}{\mathrm{d}t} = -\mathbf{u} + \mathbf{f}(\mathbf{h}) + \mathbf{W}\,\mathbf{r} + \boldsymbol{\eta} \tag{1}$$

where $\boldsymbol{\tau}$ collects the neurons' time constants, $\mathbf{u}$ denotes their membrane potentials, $\mathbf{W}$ is the synaptic weight matrix, $\boldsymbol{\eta}$ represents noise, correlated across time with time constant $\tau_\eta$, as well as across neurons with covariance $\boldsymbol{\Sigma}_\eta$, $\mathbf{f}(\mathbf{h})$ represents a non-linear transformation of the external input $\mathbf{h}$,

$$f_i(h_i) = \theta_1 \ (h_i + \theta_2)^{\theta_3} \tag{2}$$

with parameters $\boldsymbol{\theta}_h = \{\theta_1, \theta_2, \theta_3\}$, and $\mathbf{r}$ denotes neural firing rates, given by a rectified expansive non-linearity,

$$r_i = k \ \lfloor u_i \rfloor_+^\gamma \tag{3}$$

with $\gamma > 1$ and $k > 0$. Note that there is no built-in upper saturation to the model, which presents potential instability problems – excitations within the network can exponentially grow unbounded through mutual and self-excitation. The network relies purely on inhibition to stabilize, which therefore restricts the parameter regime in which the network is stable. The network comprises of distinct populations of excitatory and inhibitory neurons, imposing restrictions on the polarity of the elements of $\mathbf{W}$. Specifically, the elements of each column of $\mathbf{W}$ must have the same sign (positive for excitatory neurons, negative for inhibitory neurons).

## 2.2 Theoretical motivation

We note that under the assumption of reasonable noise characteristics, it is always possible to add a new neuron to an existing stable SSN while preserving stability in a non-trivial manner. We begin by defining the newly-added neuron as a "twin" of an existing neuron. The new twin has the same incoming recurrent weights as its original counterpart. This way, the newly-added neuron is guaranteed to receive the same input as the original neuron. Second, the newly-added neuron also inherits the outgoing weights of its original counterpart, but such that all outgoing weights of both neurons are halved. Thus, disregarding noise and external input, all other neurons in the network receive the same total input from these twin neurons as they used to receive from just the original neuron before its duplication. Therefore, still disregarding noise and external input, the activity of every neurons remains unchanged, and the newly-added neuron simply repeats the activity of its original counterpart. In other words, if the network was stable before adding the new neuron, it will also be stable after having added it. We refer to this process as the "dynamics-neutral" addition of a neuron to the network. This idea forms the mathematical foundation of our proposed training method.

Formally, let neuron $n$ be the cloning candidate. The membrane potential vector $\mathbf{u}$, and thus the firing rate vector $\mathbf{r}$ now each have an additional element:

$$\mathbf{u} = \begin{bmatrix} \mathbf{u}_{\neg n} \\ u_n \end{bmatrix} \rightarrow \begin{bmatrix} \mathbf{u}_{\neg n} \\ u_n \\ u_n \end{bmatrix}, \text{ and } \mathbf{r} = \begin{bmatrix} \mathbf{r}_{\neg n} \\ r_n \end{bmatrix} \rightarrow \begin{bmatrix} \mathbf{r}_{\neg n} \\ r_n \\ r_n \end{bmatrix} \tag{4}$$

where $\neg n$ represents the indices of all neurons that are not $n$. Similarly, we split the weight matrix $\mathbf{W}$ into four blocks, $\{\mathbf{W}_{(\neg n, \neg n)}, \mathbf{W}_{(\neg n, n)}, \mathbf{W}_{(n, \neg n)}, \mathbf{W}_{(n,n)}\}$. We update the network weights according to

$$\mathbf{W} = \begin{bmatrix} \mathbf{W}_{(\neg n, \neg n)} & \mathbf{W}_{(\neg n, n)} \\ \mathbf{W}_{(n, \neg n)} & \mathbf{W}_{(n,n)} \end{bmatrix} \rightarrow \begin{bmatrix} \mathbf{W}_{(\neg n, \neg n)} & \frac{1}{2}\mathbf{W}_{(\neg n, n)} & \frac{1}{2}\mathbf{W}_{(\neg n, n)} \\ \mathbf{W}_{(n, \neg n)} & \frac{1}{2}\mathbf{W}_{(n,n)} & \frac{1}{2}\mathbf{W}_{(n,n)} \\ \mathbf{W}_{(n, \neg n)} & \frac{1}{2}\mathbf{W}_{(n,n)} & \frac{1}{2}\mathbf{W}_{(n,n)} \end{bmatrix} \tag{5}$$

The synaptic input into all neurons (the equivalent of the third term on the RHS of Eq. 1) is now given by

$$\begin{bmatrix} \mathbf{W}_{(\neg n, \neg n)} & \frac{1}{2}\mathbf{W}_{(\neg n, n)} & \frac{1}{2}\mathbf{W}_{(\neg n, n)} \\ \mathbf{W}_{(n, \neg n)} & \frac{1}{2}\mathbf{W}_{(n,n)} & \frac{1}{2}\mathbf{W}_{(n,n)} \\ \mathbf{W}_{(n, \neg n)} & \frac{1}{2}\mathbf{W}_{(n,n)} & \frac{1}{2}\mathbf{W}_{(n,n)} \end{bmatrix} \begin{bmatrix} \mathbf{r}_{\neg n} \\ r_n \\ r_n \end{bmatrix} = \begin{bmatrix} \mathbf{W}_{(\neg n, \neg n)} \, \mathbf{r}_{\neg n} + \mathbf{W}_{(\neg n, n)} \, r_n \\ \mathbf{W}_{(n, \neg n)} \, \mathbf{r}_{\neg n} + \mathbf{W}_{(n,n)} \, r_n \\ \mathbf{W}_{(n, \neg n)} \, \mathbf{r}_{\neg n} + \mathbf{W}_{(n,n)} \, r_n \end{bmatrix} \tag{6}$$

where the top rows of this vector are

$$\begin{bmatrix} \mathbf{W}_{(\neg n, \neg n)} \, \mathbf{r}_{\neg n} + \mathbf{W}_{(\neg n, n)} \, r_n \\ \mathbf{W}_{(n, \neg n)} \, \mathbf{r}_{\neg n} + \mathbf{W}_{(n,n)} \, r_n \end{bmatrix} = \mathbf{W} \, \mathbf{r} \tag{7}$$

which is the same input as that received by the previously existing neurons, before the addition of the new neuron, as claimed (cf. Eq. 1). Note that during optimization, symmetry between the twin

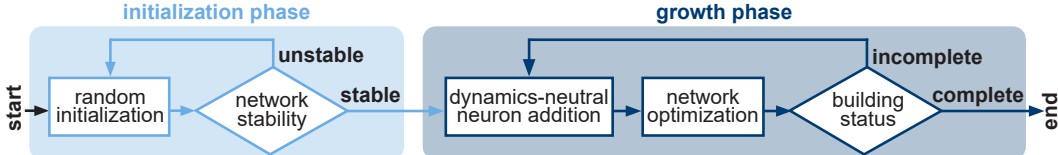

Figure 1: **Method sketch for training SSNs by dynamics-neutral growth.** As randomly initializing large SSNs yields unstable or trivially stable (silent) networks with high probability [29], we start with a small network of two neurons, comprising of a single excitatory and inhibitory neuron, by generating network parameters until a set of parameters corresponding to a stable network is found (initialization phase). We subsequently add neurons in a dynamics-neutral manner while concurrently optimizing for the training objective until the desired network size is achieved (growth phase).

neurons is broken due to them receiving statistically equal but nonetheless uniquely-generated noise inputs. As our test cases are constructed to have the same number of inputs as the number of neurons in the network, we also choose not to duplicate incoming weights from external inputs for newly added neurons, but instead to connect each newly added neuron to a single new input. (In future work, we will consider all-to-all connections between external inputs and existing neurons, with trainable weights, which are duplication at network growth.) These factors wear down the absolute guarantee of stability, but it is still reasonable to build on this intuitively high chance of stability.

## 2.3 Training method

Even with a reliable way of growing a network, we still explicitly require an existing (smaller) network as a starting point. By "backward induction", we recognize that the simplest and most reliable solution is to initialize the whole process by generating the smallest SSN possible – a network with only one excitatory and one inhibitory neuron. This can be achieved by brute-force or by some targeted initialization specific to the training objective in order to get a head-start on performance optimization. The full network can then be gradually constructed by adding neurons in a dynamics-neutral way as previously described. The entire training method is summarized in Fig. 1.

## 2.4 Training reliability

We calculate three different metrics that together characterize how reliably our method protects against instability problems typically encountered in training SSNs.

1. **Proportion of unstable networks during training.** We repeat the entire training method for a total of 100 separate networks and count the number of networks that become unstable over training (i.e. runaway activity for any of the training inputs).

2. **Proportion of unstable trials with shuffled weights.** For each of the 100 fully trained networks (see above), we shuffle the elements within each quadrant of their weight matrix, defined by the excitatory (E) or inhibitory (I) nature of the pre- and postsynaptic neurons (E-E, E-I, I-E and I-I). We repeat this shuffling 100 times, run each shuffled network for 100 trials, each with a different input (from either MNIST or CIFAR-10, see below), and count the number of trials in which the network becomes unstable. We report the mean and standard error of the proportion of unstable trials across the 10,000 shuffled networks.

3. **Proportion of unstable randomly generated networks.** For each of the 100 fully trained networks (see above), we generate 100 random networks, such that each element of their weight matrix is generated from a zero-mean Gaussian with variance equal to the variance of all weights in the corresponding trained stable network. We then set the sign of each weight to comply with the E/I identity of the presynaptic neuron. We run each random network for 100 trials with different inputs, as above, count the number of unstable trials, and report the mean and standard error of the proportion of unstable trials across the 10,000 random networks.

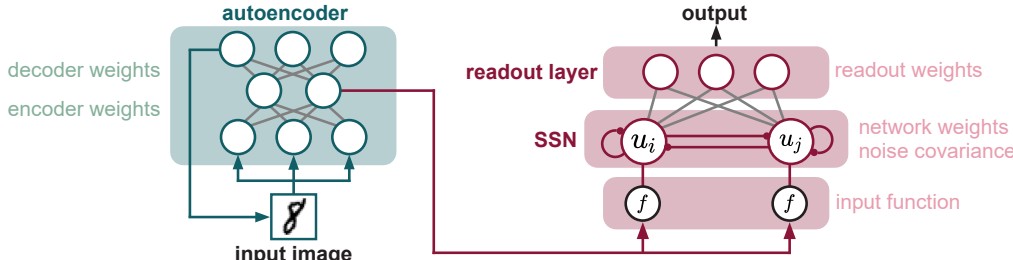

Figure 2: **Training an SSN to perform the MNIST classification task.** Right: the network consists of 40 excitatory and 10 inhibitory neurons (not shown). Left: before training the SSN, we optimize a 3-layer autoencoder (green) to reduce the dimensionality of each MNIST image to match the number of excitatory neurons in the network, receiving the encoded image as input after transformation by the input function, $f$. For each input, SSN activities, $y_i$, evolve over time until reaching their stationary distribution, after which the mean membrane potentials are passed through a feedforward readout layer to make a prediction (red). Trained parameters are highlighted on the two sides of the figure.

## 3 MNIST classification task

We first train SSNs on MNIST digit classification [33]. Due to its recurrent dynamics (Eq. 1), neural activities in an SSN evolve over time even in response to a fixed external input, instead of instantaneously producing a fixed output. Moreover, due to the presence of noise in the dynamics, a stochastic SSN never settles onto a single response. While these features are ever-present in experimental neural recordings, and thus contribute to the biological realism of stochastic SSNs, they also make their training challenging. Therefore, to define the network's output for the purposes of training (and, more generally, for measuring its performance), we evolve neural activities until they reach a steady-state distribution, and pass the mean membrane potentials into a readout (softmax) layer.

An overview of the training approach is shown in Fig. 2. We train an SSN with 80 excitatory and 20 inhibitory neurons (a biologically-realistic E:I ratio). We also train an SSN with 50 excitatory and 50 inhibitory neurons to compare with the performance of previous models [29]. Only the excitatory neurons receive external input (i.e. MNIST images) and are decoded in the readout layer, although activities in the entire network (including inhibitory neurons) evolve over time. In order to transform the MNIST images into inputs for the SSN, we optimize an autoencoder (also comprising of 3 layers) to reduce input dimensionality from 784 (number of pixels in the MNIST images) to 50 (number of excitatory neurons in the network). As a base comparison, we also train multi-layer perceptrons (MLPs) with 3 layers, and perform logistic regression on the encoded data from the autoencoder.

Training results are summarized in Table 1. None of our 200 trained networks encountered instability problems. This is highly non-trivial: at both E:I ratios, it is virtually impossible to initialize networks randomly so that they are stable, and even shuffling the weights of trained stable networks results in instability in nearly 80% of cases. In addition, our trained SSNs at both E:I ratios perform at only slightly lower levels than the corresponding MLPs, and also better than logistic regression despite having noise injected at every time step and also having to maintain dynamic stability. SSNs constrained to have a "ring" architecture (such that each E-I quadrant of the weight matrix is circulant) with Gaussian weight profiles, as in previous work [29], are not able to perform the task competently because of their highly-constrained parameterization designed to exploit rotational symmetries in their training images, which do not exist in the MNIST dataset. Nevertheless, their performance is still above chance (0.1).

## 4 Amortized probabilistic inference

We next train an SSN on a challenging probabilistic inference task. This task requires the SSN to act as the recognition model for a probabilistic generative model of natural image patches, the Gaussian scale mixture (GSM) model [34, 36], by performing amortized approximate Markov chain Monte Carlo (MCMC) inference [29] (Fig. 3). That is, in response to an input (observation), the network needs to produce trajectories in the state space of its neurons (encoding latent variables of the GSM) such that

Table 1: **Stability and performance of SSNs trained on MNIST classification**. Numbers separated by colons indicate the number of excitatory and inhibitory neurons in the SSN, respectively. The encoded data are labeled according to their reduced dimensionality (see Fig. 2. For comparison, we separately train MLPs (with 3 layers) and perform logistic regression to classify the encoded data. Despite the presence of noise, our trained SSN performs only slightly worse than the MLP and better than logistic regression.

| stability | | |
|---|---|---|
| SSN E:I ratio | networks | proportion unstable |
| 50:50 | trained networks | 0.0 |
| | networks w/ shuffled weights | 0.790 ($\pm$ 0.007) |
| | random networks | 1.0 ($\pm$ 0.0) |
| 80:20 | trained networks | 0.0 |
| | networks w/ shuffled weights | 0.794 ($\pm$ 0.008) |
| | random networks | 1.0 ($\pm$ 0.0) |
| **performance** | | |
| model | MNIST data | test accuracy |
| Logit | encoded-50 | 0.914 |
| MLP | encoded-50 | 0.974 |
| Gaussian ring SSN [29] (50:50) | encoded-50 | 0.223 |
| SSN (50:50) | encoded-50 | **0.949** |
| Logit | encoded-80 | 0.922 |
| MLP | encoded-80 | 0.976 |
| SSN (80:20) | encoded-80 | **0.952** |

the distribution of multi-neuron response patterns sampled by these trajectories approximately matches the joint posterior distribution of the GSM for corresponding input (hence performing approximate MCMC inference). Notably, the network needs to achieve this with a single set of parameters (prominently, synaptic weights) for a large number of inputs (hence the inference is "amortized"). This is a challenging task because the objective requires the network to maintain finite levels of variability while modulating the co-variability of its neurons in a stimulus-dependent way, which is only possible if stochasticity of neural noise is not suppressed and recurrent connections are sufficiently strong – a combination that pushes the network towards instability. This task has also been suggested to have neurobiological relevance. Previous work indicated that the stationary statistics of neural responses in the primary visual cortex (V1) may be accounted for by a model assuming these responses represent statistical samples from a GSM posterior [36, 37], and that SSNs producing such samples also account for other dynamical aspects of V1 responses such as oscillations and transient overshoots [29].

### 4.1 Gaussian scale mixture model

The GSM [34, 36] was proposed as a generative model of natural image patches suitable for image processing, and as an internal model that the visual system (in particular, V1) may implicitly use to process images. According to the GSM, an image $\mathbf{x}$ is generated as a linear combination of so-called "projective fields" (columns of a matrix, $\mathbf{A}$, see below), with overall scaling ("global contrast") determined by a scalar, $z$, and with some additive zero-mean, Gaussian white noise pixel noise, $\boldsymbol{\eta}_{\mathrm{x}} \sim \mathcal{N}\left(0, \sigma_{\mathrm{x}}^{2} \mathbf{I}\right)$ (where $\mathbf{I}$ is the identity matrix):

$$\mathbf{x} = z \mathbf{A} \mathbf{y} + \boldsymbol{\eta}_{\mathrm{x}} \tag{8}$$

where $\mathbf{y}$ are latent variables scaling the individual contribution of each projective field in $\mathbf{A}$. The projective fields are 2D Gabor filters, each characterized by four parameters determining the orientation, size, and 2D location of the filter (see also Appendix A in the Supplementary material) such that each column of $\mathbf{A}$ corresponds to the pixels of a single vectorized 2D Gabor filter. An ideal observer, when presented with an image $\mathbf{x}$, computes a posterior distribution over $\mathbf{y}$ and $z$ by incorporating their respective priors:

$$\mathbf{y} \sim \mathcal{N}(\mathbf{y}; 0, \mathbf{C}), \text{ and } z \sim \mathrm{Gamma}(\alpha, \beta) \tag{9}$$

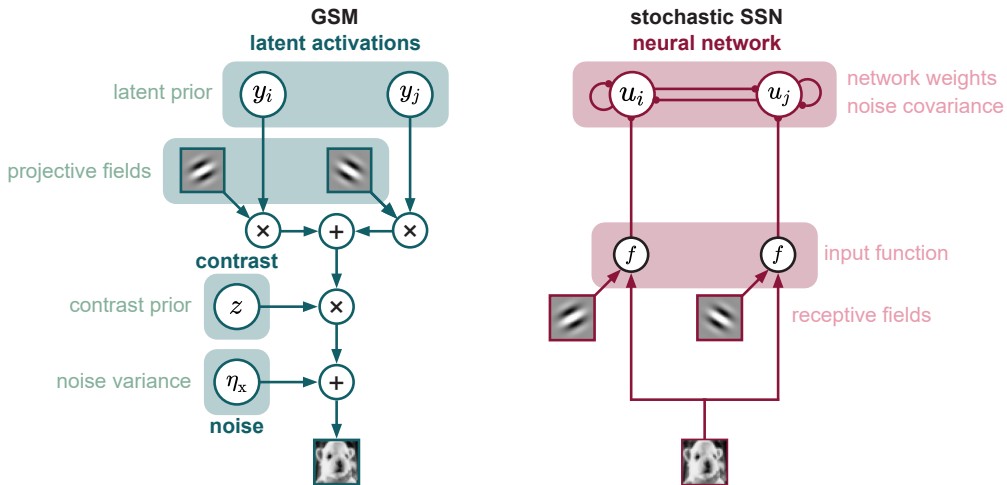

Figure 3: **Training an SSN to perform inference under a GSM.** The GSM (left) is a generative model of natural image patches proposed to underlie computations in the visual cortex. According to it, an image is constructed as a linear combination of (oriented Gabor filter-based) projective fields, each contributing according to its corresponding latent "activation", $y_i$, scaled by a global scalar contrast level, $z$, and corrupted by white zero-mean Gaussian pixel noise, $\eta_\text{x}$. We train an SSN to perform approximate amortized MCMC inference under the GSM (right). Input the the SSN is an image patch linearly filtered by oriented receptive fields (taken to be identical to the projective fields of the GSM), elementwise transformed by a nonlinear "input function", $f$. Excitatory neurons in the SSN are taken to correspond to latent variables of the GSM, such that their responses, $u_i$, given some input image, represents statistical samples from the posterior over the latent variables of the GSM given the same observed image. Trained parameters are highlighted on the two sides of the figure.

so that (by Bayes' rule)

$$\text{P}(\mathbf{y}, z | \mathbf{x}) \propto \mathcal{N}\left(\mathbf{x}; z\,\mathbf{A}\,\mathbf{y}, \sigma_\text{x}^2\,\mathbf{I}\right) \mathcal{N}(\mathbf{y}; 0, \mathbf{C})\,\text{Gamma}(z; \alpha, \beta) \tag{10}$$

after which the contrast level $z$ is marginalized, resulting in a posterior distribution over $\mathbf{y}$:

$$\text{P}(\mathbf{y} | \mathbf{x}) = \int \text{P}(\mathbf{y}, z | \mathbf{x})\,\text{d}z \tag{11}$$

We construct our GSM by first maximizing the fraction of variance explained by the projective fields on CIFAR-10 images and then optimizing its remaining parameters by maximizing model (marginal) likelihood on the same images. Additional details can be found in Appendix A.

## 4.2 Sampling-based probabilistic inference

Given an observed image as input, the membrane potential responses of the SSN, $\mathbf{u}$, as determined by Eq. 1, are required to (approximately) represent statistical samples from the posterior distribution over the latent variables of the GSM as computed by an ideal observer using Eq. 11 (assuming a one-to-one correspondence between the excitatory neurons of the SSN and the latent variables of the GSM). This inference method is therefore described as "sampling-based". We achieve this by optimizing the SSN parameters with respect to the mean-squared errors (averaged over 50,000 training images) between the mean and (co)variance of its response distributions, $\mathbb{E}_\text{SSN}[\mathbf{u}]$, $\mathbb{V}_\text{SSN}[\mathbf{u}]$, and $\mathbb{C}_\text{SSN}[\mathbf{u}]$, and of the corresponding target posterior distributions, $\boldsymbol{\mu}_\text{GSM}$, $\boldsymbol{\sigma}_\text{GSM}^2$, and $\boldsymbol{\Sigma}_\text{GSM}$:

$$\mathcal{L} = \lambda_\mu\,\|\mathbb{E}_\text{SSN}[\mathbf{u}] - \boldsymbol{\mu}_\text{GSM}\|_2^2 + \lambda_{\sigma^2}\,\left\|\mathbb{V}_\text{SSN}[\mathbf{u}] - \boldsymbol{\sigma}_\text{GSM}^2\right\|_2^2 + \lambda_\Sigma\,\|\mathbb{C}_\text{SSN}[\mathbf{u}] - \boldsymbol{\Sigma}_\text{GSM}\|_\text{F}^2 \tag{12}$$

for constant coefficients $\lambda_\mu$, $\lambda_{\sigma^2}$ and $\lambda_\Sigma$ (and where $\|\cdot\|_\text{F}$ denotes the Frobenius norm). In this case, both excitatory and inhibitory neurons receive external input corresponding to their respective receptive fields (such that an E-I pair shares the same receptive field) but, as before, only excitatory neurons are considered in the computation of the cost function.

Table 2: **Stability and performance of SSNs trained on probabilistic inference under the GSM**. SSNs have been previously been trained for the same task [29], which serves as a comparison. We train two SSNs, one for the ring GSM in [29] (labeled as ring SSN), and another for our GSM which consists of a diverse set of basis functions optimized from CIFAR-10 images (general SSN). Since training targets are different for the two GSMs, we compute a modified cost (Eq. 13) in order to directly compare optimization performance.

| stability | | |
|---|---|---|
| model | networks | proportion unstable |
| general SSN | trained networks | 0.0 |
| | networks w/ shuffled weights | $0.446 \, (\pm \, 0.008)$ |
| | random networks | $1.0 \, (\pm \, 0.0)$ |
| **performance** | | |
| network | GSM target | normalized cost |
| Gaussian ring SSN [29] | ring [29] | 1.0 |
| ring SSN | ring [29] | **0.158** |
| Gaussian ring SSN [29] | optimized for CIFAR-10 | $\sim 10^4$ |
| general SSN | optimized for CIFAR-10 | **0.745** |

## 4.3 Results

We train an SSN with circulant quadrants in the weight matrix ("ring SSN") for a GSM whose projective fields only differ in their orientation (and thus form a ring topology), used in earlier work [29]. We also train an unconstrained SSN ("general SSN") for the GSM with a richer set of projective fields we described above (Section 4.1. For both GSMs, we also train ring SSNs that are constrained to have Gaussian weight profiles (Gaussian ring), as in previous work [29]. As target moments are different in the two GSMs, we compute a normalized cost function so that we can compare networks trained across different GSMs:

$$\mathcal{L}' = \lambda_\mu \frac{\left\| \mathbb{E}_{\text{SSN}}[\mathbf{u}] - \boldsymbol{\mu}_{\text{GSM}} \right\|_2^2}{\left\| \boldsymbol{\mu}_{\text{GSM}} \right\|_2^2} + \lambda_{\sigma^2} \frac{\left\| \mathbb{V}_{\text{SSN}}[\mathbf{u}] - \boldsymbol{\sigma}_{\text{GSM}}^2 \right\|_2^2}{\left\| \boldsymbol{\sigma}_{\text{GSM}}^2 \right\|_2^2} + \lambda_\Sigma \frac{\left\| \mathbb{C}_{\text{SSN}}[\mathbf{u}] - \boldsymbol{\Sigma}_{\text{GSM}} \right\|_F^2}{\left\| \boldsymbol{\Sigma}_{\text{GSM}} \right\|_F^2} \quad (13)$$

In order to fully appreciate these results, we briefly describe the GSM-SSN pairs in each row of Table 2 and how they are trained. A full comparison can be found in Appendix B.

1. The first row describes the Gaussian ring-SSN from [29] optimized for the ring GSM from the same study. This SSN is highly constrained; for example, the entire $100 \times 100$ weight matrix is parameterized by only 8 parameters. The same 5 rotationally-symmetric images are used for both training and testing, due to the high computational costs associated with the method. Optimization is first done using Adam for a small number of iterations, followed by a zero-variance but biased semi-analytical mean-field method described in [38] (and Appendix B). The normalized cost of this model is set to 1 by definition.

2. The ring SSN in the second row is trained by our method, with the same 5 images used as the training and test set (as above). The weight matrix is constrained to be block-symmetric-circulant in order to fully exploit the rotational symmetry of the images. Each symmetric-circulant $50 \times 50$ block is parameterized by 26 parameters, giving a total of 104 parameters for the weight matrix. The additional degrees of freedom has resulted in nearly an order of magnitude lower cost.

3. The third row shows the Gaussian ring SSN from [29] trained for our GSM. Due to its highly-constrained parameterization (see above) and built-in rotational symmetry assumptions, the model is not able to succeed in matching the moments of the GSM posterior moments when observing CIFAR-10 natural images and fails to converge during training.

4. The general SSN in the fourth row has all 10,000 parameters of its $100 \times 100$ weight matrix individually trained on our GSM. Despite the increase in difficulty of the task (matching a more complex GSM), our method produces a network that achieves a normalized cost lower than that achieved by a Gaussian ring SSN specifically designed to work for a ring GSM [29].

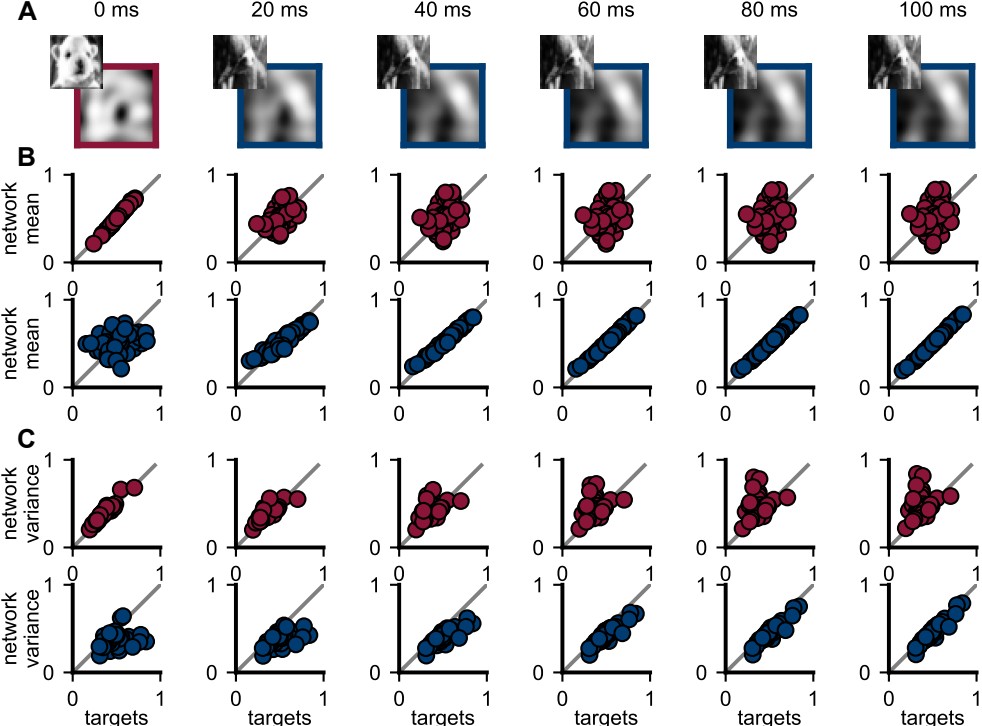

Figure 4: **Response statistics in the trained network rapidly adapt after a stimulus change.** The network is initially in steady state, with responses producing (approximate) samples from the posterior corresponding to some stimulus (red) as input. At $t = 0$ ms, a new stimulus (blue) replaces the old one as the input to the network. As a result, network responses evolve over time to represent the new posterior distribution. Each column represents the state of the network at some time $t$ after new stimulus onset (top). **A.** Input image into the network (top left) and current image "perceived" by the network, as reconstructed from the neural responses (bottom right). **B-C.** Membrane potential means (**B**) and variances (**C**) of excitatory neurons (dots) versus target GSM posterior means corresponding the old (blue) and new (red) stimuli, respectively.

In summary, our method provides two advantages. First, it allows training networks with a large number of free parameters, which in turn achieves a much lower cost than the more highly constrained networks to which previous training methods were restricted (first two rows). Second, our stability-focused approach successfully trains networks to perform inference under a complex GSM of natural images, a feat which was previously impossible (last two rows). Once again, in all 100 independent training attempts, not a single network became unstable. Full-sized stable networks are also impossible to obtain by random generation, further justifying our approach.

### 4.4 Analysis of the trained network

It is interesting to analyze the trained SSN and explore its dynamics. For this, we reverse-engineer the network in order to obtain the image that it "perceives" at a given time by interpreting the mean membrane potentials of its excitatory neurons as the latent activations of the GSM, from which an image is reconstructed using the GSM's projective fields and generative process (Eq. 8). We observe the phenomenon of percept morphing (Fig. 4A), driven by membrane potential moments initially matching the GSM targets corresponding to the old stimulus, and gradually evolving over time to instead match those corresponding to the new stimulus (Fig. 4B-C). While the target means are matched with high precision and within ∼40 ms (Fig. 4B), variances are slightly less precisely matched and take longer to converge (Fig. 4C), demonstrating the difficulty of matching higher-order moments. Furthermore, membrane potential means transition between the two posteriors faster than membrane potential variances do. From an MCMC perspective, this suggests a difference in mixing times for the two quantities.

## 5  Discussion

Network stability is a problem faced in both artificial [39, 40] and biological neural networks [17]. Typically, artificial neural networks (ANNs) can implement quick-fix solutions that will not impact their performance negatively, such as applying batch and layer normalizations [41–43]. Furthermore, artificial neurons may be designed to be stable, e.g. by simply choosing a saturating activation function (tanh, sigmoid) or by implementing other bounded-input-bounded-output designs.

In general, the concept of growing networks during training has recently been proposed in ANNs. Previous work combined both network growing and pruning to produce efficient networks [31], allowed the splitting of an existing neuron into multiple neurons [30], and used principles similar to ours for adding nodes without impacting the operation of the existing network [32]. Broadly speaking, the two primary goals of these algorithms were to reduce training load and improve performance. Here, given its particular relevance for biological networks, we instead focused on the even more basic requirement of staying within a stable dynamical regime.

Building a network gradually during optimization may adversely afftect the speed and scalability of our algorithm because it requires computationally expensive function retracing operations every time the network grows and thus changes in architecture. We justify this with two reasons. First, there is presently no other effective way of training SSNs. Second, the computational resources spent on optimization are still much greater than those spent on tracing, especially as there is only a finite number of tracing operations required (until the network reaches full size), but a much larger number of gradient computations and optimization iterations (which may need to continue even once the network reaches full size). Better balancing network building and its associated computational costs is an important future direction.

Our method may be further improved by fine-tuning its hyperparameters, such as the number of neurons to duplicate at the same time, or the criteria for selecting which neuron(s) to duplicate. In addition, our method may also be combined with methods based on fundamentally different approaches, such as adding regularization terms to the cost function explicitly encouraging networks to stay within a stable regime [27].

Throughout our analysis, we use backpropagation through time with Adam [44] to train SSNs (see Appendix C for additional information on the training procedure). This is because our primary objective is to build biologically-realistic networks so as to study their dynamics and network characteristics after they have been optimized. Future work will need to study how biologically plausible plasticity mechanisms may achieve the same goal.

Finally, as an example of a computationally particularly challenging and (for V1) biologically relevant task, we chose to train SSNs to perform sampling-based inference [29]. Our method is readily applicable to training SSNs on other cognitive tasks engaging a number of different cortical areas, including the prefrontal cortex [16], opening the way to studying SSNs beyond primary sensory cortices.

## 6  Conclusion

We present an effective method for training SSNs, a canonical model of cortical circuits highly prone to exhibit dynamic instabilities. To demonstrate the generality of our method, we train SSNs to perform MNIST image classification. We also train SSNs to perform sampling-based inference under a GSM optimized for CIFAR-10 images, a previously unattainable task. Our contribution also makes it possible to train SSNs on other neurobiologically relevant cognitive tasks and thus study the network dynamics that underlie the complex computations performed by the brain.

## Acknowledgments and Disclosure of Funding

This work was supported by the Wellcome Trust (Investigator Award in Science 212262/Z/18/Z to M.L.) and the Human Frontiers Science Program (Research Grant RGP0044/2018 to M.L.).

### Author contributions

W.S. developed the algorithm and performed the experiments. Both authors designed the study, interpreted the results, and wrote the paper.

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
