# Training stochastic stabilized supralinear networks by dynamics-neutral growth

## Supplementary material

## A  Gaussian scale mixture model

For completeness, we restate the main equation of the Gaussian scale mixture model (GSM) here. The GSM [1, 2] is a generative model of images $\mathbf{x}$ described by

$$\mathbf{x} = z\,\mathbf{A}\,\mathbf{y} + \boldsymbol{\eta}_{\mathrm{x}} \tag{1}$$

for projective fields $\mathbf{A}$ (with parameters $\theta_A$), latent activations $\mathbf{y}$ (with prior covariance $\mathbf{C}$), contrast level $z$ (with gamma prior parameters $\alpha$ and $\beta$) and pixel noise $\boldsymbol{\eta}_{\mathrm{x}}$ (with variance $\sigma_{\mathrm{x}}$). We first aim to tune the projective fields so that they are able to maximally account for the pixel strength variability of CIFAR-10 images. For some image $\mathbf{x}_i$, the least-squares reconstruction of the image $\tilde{\mathbf{x}}_i$ using only projective fields $\mathbf{A}$ is

$$\tilde{\mathbf{x}}_i = \mathbf{A}\,(\mathbf{A}^{\mathrm{T}}\mathbf{A})^{-1}\,\mathbf{A}^{\mathrm{T}}\,\mathbf{x}_i \tag{2}$$

Let images be indexed by $i$ and the pixel within each image be indexed by $j$. We seek to reduce the fraction of variance (across pixels) unexplained averaged across all images:

$$\hat{\theta}_A = \arg\min_{\theta_A} \mathbb{E}_i\left[\frac{\mathbb{V}_j[x_j - \tilde{x}_j]}{\mathbb{V}_j[x_j]}\right] \tag{3}$$

The optimized set of filters depict a diverse range of orientations, positions and scales (Fig S1A). With the projective fields optimized and now fixed, we optimize the rest of the parameters $\theta_{\mathrm{GSM}}$ by maximizing the likelihood of observing these training images under the generative model. The joint probability of observing image $\mathbf{x}_i$ at contrast level $z$ is given by

$$p(\mathbf{x}_i, z) \propto \mathcal{N}(\mathbf{x}_i; 0, z^2\mathbf{A}\,\mathbf{C}\,\mathbf{A}^{\mathrm{T}} + \sigma_{\mathrm{x}}^2\,\mathbf{I})\,\mathrm{Gamma}(z; \alpha, \beta) \tag{4}$$

We therefore minimize the negative log-likelihood summed across all images:

$$\hat{\theta}_{\mathrm{GSM}} = \arg\min_{\theta_{\mathrm{GSM}}} \left(-\sum_{i=1}^{N_i} \log \int p(\mathbf{x}_i, z)\,dz\right) \tag{5}$$

for total number of images $N_i$. On average, across all training images in CIFAR-10 [3], the optimized GSM is able to achieve $21.83\%$ variance unexplained (Fig S1C).

## B  Past and present training approaches of GSM-SSN

### B.1  Gaussian scale mixture model

We first review the differences in the GSM used in Ref. 4 and the GSM that we constructed. The projective fields of our GSM is obtained by optimizing for the fraction of variance explained on CIFAR-10 images (Fig S1A). The rest of the parameters are optimized by maximum likelihood, and the result is a realistic generative model of natural images. In the GSM of Ref. 4, Gabor filters were artificially constructed (Fig S1B) and are unable to fully capture any statistics of natural images. Furthermore, all other parameters were chosen by hand.

### B.2  Training images

Since our GSM is optimized for CIFAR-10, we simply partition the dataset for training and testing. In Ref. 4, 5 images were constructed using the basis functions from its own GSM. These same 5 images were used in both testing and training.

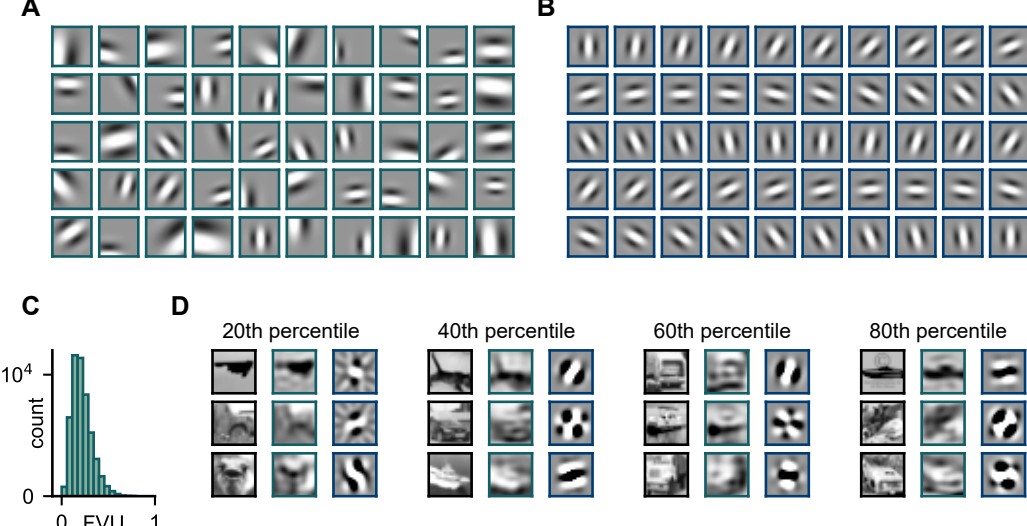

Figure S1: **Comparison of GSMs. A.** Projective fields of the GSM after optimizing for fraction of variance unexplained (FVU) on CIFAR-10 images. The filters span a diverse range of orientations, positions and scales in order to capture the statistics of natural images. **B.** GSM projective fields in a previous approach. The projective fields are generated by Gabor filters of the same frequency and position, differing only in their orientations [4]. These filters are also the receptive fields of SSNs trained in past approaches. The rotational symmetry of the filters allow for low-dimensional parameterizations of the corresponding SSN, such as rotationally-symmetric weight and noise matrices. **C.** Histogram of computed FVUs of all 50000 images in the test set. **D.** Examples of least-squares reconstructed images in different percentiles of FVUs using filters from (**A**) in the middle columns and (**B**) in the right columns. Original images are in the left columns.

## B.3   SSN parameterizations

As mentioned in the main text, we trained two networks, one on the GSM of Ref. 4 ("ring SSN") and another on our own GSM ("general SSN").

**Input function.** For both of our networks and in Ref. 4, the input function is parameterized by 3 parameters, as stated in the main text, which we will repeat here:

$$f_i(h_i) = \theta_1 \ (h_i + \theta_2)^{\theta_3} \tag{6}$$

for parameters $\theta_h = \{\theta_1, \theta_2, \theta_3\}$.

**Weight matrix.** In all networks, there are 50 excitatory and 50 inhibitory neurons, resulting in a weight matrix of size $100 \times 100$. This corresponds to 10000 free parameters in the general SSN. For Ref. 4 and the ring SSN, we partition the weight matrix into four $50 \times 50$ blocks, corresponding to all E-E, E-I, I-E and I-I connections respectively. Due to the rotational symmetry of the projective fields in these two networks, we may constrain each block to be symmetric circulant, and therefore each block is fully determined by the first half-row of the block. This corresponds to 26 parameters per block (104 in total) for the ring SSN. For Ref. 4, each block is only parameterized by two terms, corresponding to the height and width of a circular Gaussian curve (8 parameters in total).

**Noise matrix.** The noise matrix determines the covariance structure of the noise, and is therefore also a $100 \times 100$ matrix. The parameterizations are exactly the same as the weight matrix for the ring SSN (104 parameters) and the general SSN (10000 parameters). In Ref. 4, only 4 parameters are used.

**Summary.** For Ref. 4, there are a total of

$$3 \text{ (input)} + 8 \text{ (weights)} + 4 \text{ (noise)} = 15 \text{ parameters}$$

For the ring SSN, there are

$$3 \text{ (input)} + 104 \text{ (weights)} + 104 \text{ (noise)} = 211 \text{ parameters}$$

and for the general SSN:

$$3 \text{ (input)} + 10000 \text{ (weights)} + 10000 \text{ (noise)} = 20003 \text{ parameters}$$

### B.4 Training algorithm

For the ring and general SSNs, we initialize membrane potentials at resting value (defined to be 0mV), and then add a constant external stimulus dependent input. We allow the network to evolve over 500ms in 100 parallel trials, after which we compute the membrane potential mean $\mathbb{E}_{\text{SSN}}[\mathbf{u}]$ and covariance $\mathbb{C}_{\text{SSN}}[\mathbf{u}]$ across all trials and average them over an additional 100ms. These are the network moments to be compared with target moments in the cost function, which we also restate here:

$$\mathcal{L} = \lambda_\mu \left\| \mathbb{E}_{\text{SSN}}[\mathbf{u}] - \boldsymbol{\mu}_{\text{GSM}} \right\|_2^2 + \lambda_{\sigma^2} \left\| \mathbb{V}_{\text{SSN}}[\mathbf{u}] - \boldsymbol{\sigma}^2_{\text{GSM}} \right\|_2^2 + \lambda_\Sigma \left\| \mathbb{C}_{\text{SSN}}[\mathbf{u}] - \boldsymbol{\Sigma}_{\text{GSM}} \right\|_F^2 \tag{7}$$

after which we minimize this cost function by backpropagation through time using Adam.

### B.5 Semi-analytical approach

The SSN in Ref. 4 was mainly trained using a semi-analytical method. By assuming that the membrane potentials are characterized by a Gaussian process, it is possible to compute the time evolution of membrane potential means and covariances over time, as described in equations (33)-(37) of Ref. 5. This means that networks were not actually simulated. This semi-analytical result was then compared against the target moments, and the parameters of the network were optimized using the L-BFGS algorithm.

## C  Additional training details

The code to all our models can be found at

https://github.com/wmws2/stableSSN

All experiments were run on our high-performance computing cluster consisting of 4 A100 (80GB) GPUs. We have used CIFAR-10 [3] and MNIST [6] for training and testing our models. To the best of our knowledge, we have not encountered any personally identifiable information or offensive content within the datasets. The exact initialization of every parameter can be found in Table S1. Every optimization is performed using Adam [7] with default parameters.

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

Table S1: **Summary of model parameters**.

| MLP (MNIST) | | |
| --- | --- | --- |
| description | size | type |
| weights (input $\rightarrow$ layer 1) | $N_{\text{exc}} \times 360$ | ReLU |
| weights (layer 1 $\rightarrow$ 2) | $360 \times 120$ | ReLU |
| weights (layer 2 $\rightarrow$ prediction) | $120 \times 10$ | softmax |

| Autoencoder | | |
| --- | --- | --- |
| description | size | type |
| encoding weights (input $\rightarrow$ layer 1) | $N_{\text{exc}} \times 360$ | ReLU |
| encoding weights (layer 1 $\rightarrow$ 2) | $360 \times 120$ | ReLU |
| encoding weights (layer 2 $\rightarrow$ code) | $120 \times 40$ | sigmoid |
| decoding weights (code $\rightarrow$ 1) | $40 \times 120$ | ReLU |
| decoding weights (layer 1 $\rightarrow$ 2) | $120 \times 360$ | ReLU |
| decoding weights (layer 2 $\rightarrow$ output) | $360 \times N_{\text{exc}}$ | sigmoid |

| GSM | | | |
| --- | --- | --- | --- |
| parameter | description | initialization | optimized |
| $\phi$ | orientation of a Gabor filter | $\mathcal{U}(\phi; -\pi, \pi)$ | yes |
| $x_0, y_0$ | pixel coordinates of a Gabor filter | $\mathcal{U}(x_0; -16, 16)$ | yes |
| $\sigma$ | scale parameter of a Gabor filter | $\mathcal{U}(\sigma; 0.2, 1.0)$ | yes |
| $\sigma_x^2$ | pixel noise variance | 1.0 | yes |
| $\mathbf{C}$ | prior latent covariance | least-squares fit | yes |
| $\alpha$ | shape parameter of contrast prior | 2.0 | yes |
| $\beta$ | rate parameter of contrast prior | 0.5 | yes |

| SSN | | | |
| --- | --- | --- | --- |
| parameter | description | initialization | optimized |
| $\theta_1$ | constant in input function | $\mathcal{N}(\theta_1; 0, 1)$ | yes |
| $\theta_2$ | baseline in input function | $\mathcal{N}(\theta_2; 0, 1)$ | yes |
| $\theta_3$ | exponent in input function | $\mathcal{N}(\theta_3; 0, 1)$ | yes |
| $\mathbf{W}$ | weight matrix | $\mathcal{N}(w_{ij}; 0, N^{-1})$ | yes |
| $\mathbf{N}$ | noise matrix | $\begin{pmatrix} 1 & 0.1 \\ 0.1 & 1 \end{pmatrix}$ | yes |
| $\mathbf{A}$ | neuron receptive fields | GSM projective fields | no |
| $k$ | constant in firing-rate non-linearity | 0.3 | no |
| $\gamma$ | exponent in firing-rate non-linearity | 2.0 | no |
| $\tau_e$ | time constant of excitatory neurons | 20ms | no |
| $\tau_i$ | time constant of inhibitory neurons | 10ms | no |
| $\tau_\eta$ | time constant of noise | 20ms | no |
| $T$ | total simulation time | 500ms | no |
| $\lambda_\mu$ | coefficient of mean term in cost | $0.01\ N^{-1}$ | no |
| $\lambda_{\sigma^2}$ | coefficient of variance term in cost | $0.02\ N^{-1}$ | no |
| $\lambda_\Sigma$ | coefficient of covariance term in cost | $0.01\ N^{-2}$ | no |

| Adam | | | |
| --- | --- | --- | --- |
| parameter | description | initialization | optimized |
| beta_1 | first moment exponential decay rate | 0.9 | no |
| beta_2 | second moment exponential decay rate | 0.999 | no |
| epsilon | second moment exponential decay rate | $10^{-7}$ | no |