# OpenReview forum: "Training stochastic stabilized supralinear networks by dynamics-neutral growth"
_NeurIPS.cc/2022/Conference — NeurIPS 2022 Accept_

### Official Review · Reviewer_rMNf · 2022-06-19

**Rating:** 5
**Confidence:** 4
**Soundness:** 2 fair
**Presentation:** 2 fair
**Contribution:** 3 good

**Summary:**

Stabilized supralinear networks (SSN) offer a biologically plausible model of cortical neural circuits, however, previous works have had difficulty training these models in a stable way due to the model's construction. This work focuses on how to effectively train these models using gradient-based techniques and relatively few constraints, specifically in application to the task of using SSNs as a recognition model that performs probabilistic inference under a Gaussian scale mixture (GSM) generative model.  The training method consists of initially starting with a small SSN that is easier to initialize stably, and then gradually building the network up during optimization. An iterative algorithm is used where the randomly initialized network is simulated to check for convergence. If it converges, statistics are collected and used in a differentiable loss function to generate a new set of parameters. After a certain number of iterations, more neurons are carefully added to the network. The procedure repeats until the network is fully built out and the optimization of parameters is complete.

The results of the new method are compared to the existing setup in the previous work [35], and are shown to significantly reduce the cost. The network is also used to perform inference in a GSM optimized for CIFAR-10.



**Questions:**

1. The paper would benefit from a more thorough explanation of the first experiment in section 4.1 and the "existing setup in [35]" referenced in line 209. What is the training and test data? The training size is 5? What is the biased semi-analytical method used by the previous method? What are the previous method's free parameters?  A few related details are sprinkled throughout the text (lines 80-81, 146-147), however, a consolidated high-level description in Section 4.1 (and perhaps a more detailed description in the appendix) would help the reader better understand the comparison and superiority claims.

2. Partly related to the point above, I do not understand the evaluation method used for the CIFAR-10 experiment. It seems the previous method was not evaluated on CIFAR-10, and instead some kind of correction is applied? Why was the previous method not just evaluated on CIFAR-10 also? Perhaps it is not feasible to train the previous method on the larger dataset? If the latter is the case, stating that explicitly would be helpful. What should I take away from comparing the different methods with different targets and why is this valid? Is 0.788 vs 1 a significant improvement? These are questions I have that currently make it hard to evaluate the claims related to this experiment.

3.  I think the point of the overfitting discussion in Section 4.2 is that it is an accomplishment just to be able to train this relatively unconstrained model and get good performance on the training set, and the overfitting can be easily handled with regularization. But why not just directly show this by applying some of the referenced regularization methods? Is there something about the current setup that makes this nontrivial?

4. Generality: The entire paper is geared around training SSNs as a recognition network for a GSM model. A general reader may come away from the paper wondering if this is the only way this method can be used? The answer appears to be no and the authors mention their desire to apply the approach to cognitive tasks as in [16] for future work. But the current organization of the paper makes it unclear how well this method can be adapted to these other problems. I would perhaps recommend presenting the general approach first outside of the context of the recognition network setting, and then show how the probabilistic inference method can be used in the context of the recognition network for the GSM model and any specifics related to that. Ideally, there might also be another experiment, even if simple or synthetic, where this approach is used for a different setup. This is just a suggestion that I think could strengthen the presentation.

Minor points:

- Typos in lines 95-96 ("is neural noise is")  lines 123 ("the some")
- I think a better explanation of coding vs non-coding neurons in Section 2.3 would help readers less familiar with these models.
- I would recommend using a different letter than $n$ for the variable in equation 10 and line 126. This same letter $n$ is later used to refer to specific neurons in section 3.3, and this could lead to confusion for the $n=2$ point in lines 181.
- I think parenthesis are missing in Equation 16? It seems like it should be $\mathbb{E}[u] - (\mathbb{E}[f(h)] + W\mathbb{E}[r])$.






**Limitations:**

The authors adequately discuss limitations such as concerns around how well this method will scale computationally to larger problems.

**Strengths And Weaknesses:**

- Originality: The idea to use the evolutionary approach to gradually build the network up to avoid stability problems appears to be novel for SSNs. The approach of interleaving gradient optimization steps with the evolutionary steps also appears to be original. In addition, this appears to be the first practical method to train SSNs without the constraints previous approaches required. However, the paper would benefit from a more detailed discussion of evolutionary approaches such as this (mentioned briefly in lines 246-248) as well as the previous attempts to train SSNs (mentioned briefly in the introduction) to help the reader understand the literature landscape and the specific contributions of this paper.

- Quality: The general approach and mathematical details appear to be sound. The approach seems to outperform the previous method on the existing task (though see Questions section below for concerns about the explanation of this setup).  The approach was also used for a more challenging task of optimizing the GSM on CIFAR-10 images. However, clarity issues related to the explanation of the comparison for this task currently make the claims of improved performance on this task hard to evaluate (See detailed questions concerning this below).

- Clarity: The paper is generally written well, but various typos in the text and equations do make it less smooth to read at times. See the questions section below for some specific examples, but thorough proofreading would help the presentation. In addition, as mentioned in the Quality section above (and in more detail in the Questions section below), there seem to be important details related to the experiment setup that are missing which makes it confusing and hard to evaluate the empirical claims.

- Significance: The ability to effectively train SSNs at scale could have an important impact on the computational neuroscience community. I think this paper makes a step in that direction. The ideas of incorporating gradient-based optimization along with potential topological changes during training could be of interest to the broader machine learning community.

---

> ### Author Response · Authors · 2022-08-02
> **Response to reviewer rMNf**
>
> > However, the paper would benefit from a more detailed discussion of evolutionary approaches such as this (mentioned briefly in lines 246-248) as well as the previous attempts to train SSNs (mentioned briefly in the introduction) to help the reader understand the literature landscape and the specific contributions of this paper.
>
> We thank the reviewer for the comments. We have expanded on the discussion of neural networks that grow in lines 263-270 of the revised submission.
>
> > ... clarity issues related to the explanation of the comparison for this task currently make the claims of improved performance on this task hard to evaluate (See detailed questions concerning this below).
>
> > In addition, as mentioned in the Quality section above (and in more detail in the Questions section below), there seem to be important details related to the experiment setup that are missing which makes it confusing and hard to evaluate the empirical claims.
>
> We apologize for the confusion regarding details about previous attempts, and have now consolidated all information regarding this into section 4.3, as well as a dedicated section in the supplementary (see Appendix B).
>
> > Significance: The ability to effectively train SSNs at scale could have an important impact on the computational neuroscience community. I think this paper makes a step in that direction. The ideas of incorporating gradient-based optimization along with potential topological changes during training could be of interest to the broader machine learning community.
>
> We agree with the reviewer on this point and have revised the narrative to better suit the broader machine learning community.
>
> > The paper would benefit from a more thorough explanation of the first experiment in section 4.1 and the "existing setup in [35]" referenced in line 209...
>
> As mentioned above, we have now included all the information required to evaluate the superiority claims in section 4.3 and a dedicated section in the supplementary.
>
> > Partly related to the point above, I do not understand the evaluation method used for the CIFAR-10 experiment...
>
> The reviewer is right to notice that different GSMs result in different targets, which make direct comparison difficult. We had briefly mentioned in our old submission that we performed some normalization to allow for direct comparison without further detail, which was an oversight on our part. We now include the equation for normalized cost in equation (15) of the revised submission.
>
> > ... the overfitting can be easily handled with regularization. But why not just directly show this by applying some of the referenced regularization methods? Is there something about the current setup that makes this nontrivial?
>
> We have solved the overfitting problem by training the SSN on a smaller set of 10000 images and using the remaining 40000 images as a test set. We previously did not implement the referenced regularization methods as this would make them hard to compare with past approaches.
>
> > Generality: The entire paper is geared around training SSNs as a recognition network for a GSM model. A general reader may come away from the paper wondering if this is the only way this method can be used?
>
> We fully agree with this point of view and have restructured the paper to focus on the training of SSNs via growing the network. We have also included a new section where we trained an SSN to perform the well-known MNIST classification task to address these concerns in our revised submission.
>
> > Typos in lines 95-96 ("is neural noise is") lines 123 ("the some")
>
> We apologize for the typos and have made sure that there are none in our revised submission.
>
> > I think a better explanation of coding vs non-coding neurons in Section 2.3 would help readers less familiar with these models.
>
> We apologize for the confusion. We have in fact decided to remove the term “coding” in our revised submission and now explicitly state which neurons are being used for computing the cost function in both tasks.
>
> > I would recommend using a different letter than n for the variable in equation 10 and line 126. This same letter n is later used to refer to specific neurons in section 3.3, and this could lead to confusion for the  point in lines 181.
>
> We have now replaced n with $\gamma$ for the exponent for the variable in equation (3). We thank the reviewer for pointing this out.
>
> We thank the reviewer for their constructive comments.

---

> > ### Comment · Reviewer_rMNf · 2022-08-06
> > **Thank you for your response**
> >
> > Thank you for your detailed rebuttal, it has helped answer several of my questions. I also appreciate the work you have performed to restructure the paper in light of the common reviewer feedback regarding the emphasis on the method vs application. Overall I think the new revision reads much better than before! However, I still have concerns regarding the presentation and evaluation.
> >
> > **Revised paper structure**: As I mentioned, I think the overall structure of the paper is much improved. However, one thing I noticed is the amount of space taken up by the introduction and general discussion/conclusion vs the explanation of the experiments. Given the importance of the empirical evidence for your paper and the fact that the experiments seem to be performed in non-standard settings (I recognize this is mostly because there is not a ton of prior work in this area since most previous attempts have not worked too well, so I want to be cognizant of that!), I think you would still benefit from more discussion/justification of the experiments in the main text.  As you will see in my questions below, I still find myself asking many questions regarding why the experiments are set up the way they are, and why direct comparisons with the prior methods are not performed. I admittedly read this new revision quickly, so it is possible I have missed some of these points, but it still feels like the reader shouldn't have to work so hard to understand some of these things.
> >
> > **MNIST**: Thank you for including the new MNIST experiments to show the SSN training approach you propose can work in more general settings. However, I have some concerns with the evaluation.
> >
> > 1. My first question is again why the prior approaches of training SSNs are not evaluated here? I again suspect the answer is because the prior methods may not be practical, but this should be explained here if that is the case.
> >
> > 2. An explanation of the non-standard train/val split would be helpful. You mentioned in your rebuttal that this was done for regularization purposes on Cifar-10 to allow for comparison to prior approaches that did not use regularization techniques. Is that why this was done here? It does not appear you are comparing to any prior approaches, so why not use regularization techniques and evaluate on the full dataset to allow for comparison to the many other MNIST benchmark scores?
> >
> > 3. Do the results support the claim? Is 92.8% "only slightly lower" than 94.3%? Relative to what? More baselines would be helpful. Ideally, the previous SSN training method would be evaluated (per the point above), or an explanation of why not if it is impractical. How does logistic regression perform?  This is admittedly a random blog I found during a quick search: https://atmamani.github.io/projects/ml/mnist-digits-classification-using-logistic-regression-scikit-learn/#Predicting-on-full-MNIST-database, but it seems to suggest logistic regression achieves 91.9% on the normal train/val split of MNIST.  To be clear, I am not suggesting your method needs to achieve state-of-the-art results since that is not the point of your paper. But the reader does need to have some reasonable sense of how well your training procedure performs which I do not think the current results provide.
> >
> > **GSM**: I appreciate the expanded discussion of this experiment compared to previously. However, I still have some doubts. To my understanding, the main result is that SSN_2 achieves a lower normalized loss on the larger and more complex Cifar-10 experiment than the previous method achieves on a simpler experiment. I find the inclusion of the normalized cost function helpful, but still think an expanded discussion of why this is a reasonable thing to do would be helpful. I still do not understand why the prior method was not just evaluated on Cifar-10? Is this just impractical? If so, a clear explanation of this would be helpful to understand the normalization choice.
> >
> >
> > I want to be clear that I really like the ideas and think the general approach of this paper is promising! However, given that the proposed approach is a heuristic (which in itself I do not view as a negative!), rigorous and clear experimental results are required to evaluate the method. I do not think the current version of the paper provides that. However, I am open to having my mind changed if the other reviewers think I am missing important points (I again admit I read this new revision quickly, mostly focusing on the experimental sections.)

---

> > > ### Author Response · Authors · 2022-08-06
> > > **Discussion with Reviewer rMNf**
> > >
> > > We thank the reviewer for partaking in the author-reviewer discussion. We appreciate the constructive feedback.
> > >
> > > > However, one thing I noticed is the amount of space taken up by the introduction and general discussion/conclusion vs the explanation of the experiments [...]
> > >
> > > The introduction and conclusion have been visibly shortened to free up space for addressing subsequent concerns below.
> > >
> > > > ... it still feels like the reader shouldn't have to work so hard to understand some of these things.
> > >
> > > We note the reviewer's concern. We would preface by saying that in the last revision, we have added key metrics that show the stability of our method (key issue), and placed less importance about performance on any tasks, so we seek to remedy this. We hope our latest analysis on performance below is satisfactory to the reviewer.
> > >
> > > > My first question is again why the prior approaches of training SSNs are not evaluated here?
> > >
> > > Our original goal of the MNIST experiment was to demonstrate that it is possible to train an SSN outside of the highly-niche setting of sampling-based inference under the GSM. We thank the reviewer for the suggestion and have promptly performed the experiment on previous models. We have also included the accuracy from logistic regression as another base comparison. For the MNIST experiment, we now have 3 models:
> > >
> > > - 80 exc/20 inh: to address the concern of another reviewer about the E-I ratio of our networks
> > > - 50 exc/50 inh: to be compared to previous approaches
> > > - previous approaches (also 50 exc/50 inh)
> > >
> > > > An explanation of the non-standard train/val split would be helpful [...] so why not use regularization techniques and evaluate on the full dataset to allow for comparison to the many other MNIST benchmark scores?
> > >
> > > We are not able to train an SSN with ~800 neurons due to memory limitations (even on 4x80GB A100 clusters in a high-performance computing facility). This means that the MNIST dataset, whose input has 784 elements has to be compressed with an autoencoder. At this point, the standard MNIST benchmarks do not apply to this new "compressed MNIST" dataset anymore. We have also revised our experiments to adopt the standard train/val split. Standard training images are used for training, standard test images are used for testing. (This is not mentioned in the main text as we are simply adopting standard practices -- as expected).
> > >
> > > > I still have some doubts [...] I still do not understand why the prior method was not just evaluated on Cifar-10? Is this just impractical? If so, a clear explanation of this would be helpful to understand the normalization choice.
> > >
> > > There is technically nothing absolute that prevents us from evaluating the prior method on CIFAR-10. Therefore, we went ahead and did it and have added the results to the main paper. However, we already know that its performance is going to be abysmal due to its in-built rotational symmetry assumption about its input images, as well as other forms of constraints in its parameterization. We agree with the reviewer that the inclusion of this data is crucial for exemplifying the strengths of our method, and showing the exact scale of how much superior our method is.
> > >
> > > At the same time, we have also adopted the standard train/val split here. The difference now is that we have used all images (training + testing) in our optimization of the GSM. Previously, the GSM was only optimized for the training images, and was not the maximum likelihood model for the test images (and were thus inaccurate targets). There are no overfitting issues now even when following standard train/val splits. Similar to before, this is not included in the main text as overfitting did not exist in the first place in the current narrative and standard splitting is already expected.
> > >
> > > > However, given that the proposed approach is a heuristic, rigorous and clear experimental results are required to evaluate the method.
> > >
> > > We would just like to quickly emphasize that our method is **theoretically-motivated** in terms of building a network in a stable manner (requiring a brief read of section 2.2). Most importantly, our method brings into realization (from impossibility) freely-parameterized SSN models for the neuroscience and wider machine-learning community, which we feel is a signficant contribution.
> > >
> > > All in all, we hope our latest revision addresses all the reviewer's concerns, and we deeply appreciate the reviewer for taking the time to browse through our experiments.

---

> > > > ### Comment · Reviewer_rMNf · 2022-08-07
> > > > **Thank you**
> > > >
> > > > Thank you for your response and the additional experiments. I especially appreciate the inclusion of the prior method.

---

### Official Review · Reviewer_PiTG · 2022-07-11

**Rating:** 7
**Confidence:** 3
**Soundness:** 3 good
**Presentation:** 4 excellent
**Contribution:** 3 good

**Summary:**

This paper studies stochastic stabilized supralinear networks (SSNs), a firing rate recurrent neural network model of interest in computational neuroscience. The parameters of these models are typically set by hand, at random, or using semi-analytical methods. The current work approaches the problem of learning the parameters of stochastic SSNs, aiming at developing a method that can scale to large numbers of parameters. The authors focus on the problem of learning to do amortized inference for a well-known Bayesian model for vision, by optimizing a moment-matching objective function.

**Questions:**

Two things weren't very clear to me:
- Why is the stochastic SSN particularly well-suited to (approximately) sample from a GSM?
- How were the target moments obtained, for the objective function (11)?

**Limitations:**

The discussion is sufficient for the paper.

**Strengths And Weaknesses:**

The paper is clear, well written, and pleasing to read. It does a very good job in summarily explaining the current interest in the SSN in neuroscience, and the challenges (and interests) in optimizing it with deep learning techniques.

The one weakness I see in this paper is that it is purposefully written as a somewhat "niche" paper. The problem of learning to do amortized inference on a GSM is both interesting and challenging enough (thus well chosen) for the current paper, but the exclusive focus on it may deter many from reading the paper.

I think it would be possible to write a paper that would be far more reaching within the NeurIPS community and beyond, by abstracting what I understood to be the key challenge here: training recurrent neural networks with the expansive nonlinearity of eq. 10. There has been interest throughout the years in training recurrent neural networks with unbounded nonlinearities (typically rectified linear units, see e.g. the path-SGD paper by Neyshabur et al., 2015. The present paper fits nicely along this line of work, while focusing on the even more challenging case of an expansive nonlinearity. The paper could also discuss more explicitly previous work on neural networks that grow throughout training.

A small section on the easier case of deterministic SNs could be nice too, as one could make direct contact with an exciting line of recent work on so-called deep equilibrium models, which also face stability issues, and see what the current method brings there. It seems to me that some of the techniques investigated in this line of work to remedy instability (see e.g. Bai et al, ICML 2021) could be brought directly to deterministic SNs, and perhaps adapted to stochastic SNs?

There is a small number of typos in the paper. Also, there is no confusion, but perhaps it's better to denote a minimizer by arg min instead of min? I guess the F subscript can be dropped in eq. 16, as the argument of the norm is a vector?

---

> ### Author Response · Authors · 2022-08-02
> **Response to reviewer PiTG**
>
> > The paper is clear, well written, and pleasing to read. It does a very good job in summarily explaining the current interest in the SSN in neuroscience, and the challenges (and interests) in optimizing it with deep learning techniques.
>
> We thank the reviewer for the encouraging words.
>
> > The one weakness I see in this paper is that it is purposefully written as a somewhat "niche" paper. The problem of learning to do amortized inference on a GSM is both interesting and challenging enough (thus well chosen) for the current paper, but the exclusive focus on it may deter many from reading the paper. I think it would be possible to write a paper that would be far more reaching within the NeurIPS community and beyond, by abstracting what I understood to be the key challenge here: training recurrent neural networks with the expansive nonlinearity of ...
>
> We fully agree with the insight provided by the reviewer. We have completely restructured the paper to focus more on the methodology of training SSNs and provided explanations on how the main challenge stems from the expansive non-linearity and noise dynamics in our revised submission.
>
> > The paper could also discuss more explicitly previous work on neural networks that grow throughout training.
>
> We have included a review of recent works on neural networks that grow in the discussion section, including a discussion on why our method is different. Please see lines 263-270 in our revised submission.
>
> > A small section on the easier case of deterministic SNs could be nice too, as one could make direct contact with an exciting line of recent work on so-called deep equilibrium models, which also face stability issues, and see what the current method brings there. It seems to me that some of the techniques investigated in this line of work to remedy instability (see e.g. Bai et al, ICML 2021) could be brought directly to deterministic SNs, and perhaps adapted to stochastic SNs?
>
> We thank the reviewer for the suggestion. As we have mentioned in the discussion, artificial neural networks typically avoid instability issues simply by using a saturating output function and various forms of normalizations. Deep equilibrium models are an excellent counterexample of this idea, as these “quick-fixes” are unable to solve the non-trivial instability issues encountered in training DEQs (which arise from a growing number of function evaluations) and are very worthy of mention. We have added that in our discussion section in the revised submission as such an example.
>
> > There is a small number of typos in the paper.
>
> We apologize for the typos and have reviewed the revised submission and ensured that there are no typos.
>
> >Also, there is no confusion, but perhaps it's better to denote a minimizer by arg min instead of min? I guess the F subscript can be dropped in eq. 16, as the argument of the norm is a vector
>
> We thank the reviewer for pointing them out. We have made the changes in our revised submission. We have also rewritten the cost function as a summation of elements in the vector/matrix to avoid any confusion when using norms.
>
> > Why is the stochastic SSN particularly well-suited to (approximately) sample from a GSM?
>
> The GSM, when combined with the concept of sampling-based inference, is able to account for the changes in neural variability in response to visual stimuli (Orbán et al., 2016). As it turns out, such changes in neural variability are not present in simpler models. For example, linear models are unable to modulate their response variance to visual stimuli. The SSN has been found to be able to achieve this through non-linear interactions in the dynamics (Hennequin et al., 2016). This makes the SSN a good candidate to be trained to perform sampling-based probabilistic inference under a GSM and produce this effect of variability modulation.
>
> Orbán, Gergő, et al. "Neural variability and sampling-based probabilistic representations in the visual cortex." Neuron 92.2 (2016): 530-543.
>
> Hennequin, Guillaume, et al. "The dynamical regime of sensory cortex: stable dynamics around a single stimulus-tuned attractor account for patterns of noise variability." Neuron 98.4 (2018): 846-860.
>
> > How were the target moments obtained, for the objective function (11)?
>
> The target moments are obtained by performing probabilistic inference under the GSM. That is, we observe the image (in this case, images from CIFAR-10), and compute a posterior distribution over the latent variables y. The target moments are the moments of this posterior distribution.
>
> We thank the reviewer for their helpful comments.

---

> > ### Comment · Reviewer_PiTG · 2022-08-08
> > **Thank you for the responses and changes**
> >
> > Sorry for the late reply. I've now read the discussion thread with the other reviewers as well as the original rebuttals. I'd like to thank the authors for their work, and the other reviewers for their great questions. The paper has improved substantially.
> > I still vote for acceptance for this paper.

---

### Official Review · Reviewer_2UAi · 2022-07-22

**Rating:** 5
**Confidence:** 3
**Soundness:** 3 good
**Presentation:** 3 good
**Contribution:** 2 fair

**Summary:**

This paper presents a method for training a neuro-plausible artificial neural network.  _Stochastic supralinear networks_ (SSNs) have been shown to have favourable parallels to real cortical neural networks and dynamics.  This is in-part owing to the network's flexible parameterization and reduced constraints.  However, this flexibility and lack of constraints leads SSNs to be prone to instability (from recurrent feedback loops) and difficult to train at scale, without (re-)introducing additional constraints.  The authors present a method where the size of the network is grown during training, and is able to reliably reach far larger network sizes and parameter counts than preceding methods.  The method is benchmarked on an existing and new task, and appears to perform favourably.


**Questions:**

I raise several questions in my main review.

**Limitations:**

I don’t see any particular unaddressed limitations of the work as presented.  That said, it is ostensibly an empirical study of a design heuristic, and therefore the robustness and scalability is uncertain.


**Strengths And Weaknesses:**

# Review Summary

I will preface my review by saying that I am not a domain expert.  While I have a feel for neuro-plausible networks, I am not too familiar with the general architecture or existing work.

The paper itself is fairly well written, seems to be thoroughly referenced, and there are helpful figures and diagrams that help to take some explanatory burden off of the text.  The work also appears theoretically sound.  The model/architecture itself is clearly and succinctly described.  The description of the method is a little lacking, and the description of the experimental setup and objectives is also a little lacking.  I do think a method like this could be well received, and would definitely have an audience at NeurIPS.

That said, I am not going to endorse publication at this stage.  The thrust of my criticisms are (1) the foundations of the method, (2) some aspects of the presentation, and (3) the evaluation.


# Major Comments:

### 1.  The Method Itself
The core concept of the method is that it is easier to grow a stable network into a slightly larger stable network, than to outright train a large and stable network.  However, this is essentially a heuristic.  It is not clear to me that this is a robust method for generating a network.  There are two phases to the method:  initialisation and growth.  The initialisation is essentially trial-and-error in the base case of the “induction”.  The growth phase is then hoping to expand the network capacity in such a small step that it remains stable – but it is also a heuristic, that is essentially trial-and-error, in which the chance of “error” is minimised.  It is appealingly simple, but it lacks the rigour for me to be super excited / confident in the methodology.

There is also little exploration of "hyper parameters"/choices, such as how frequently the network should be grown, further heuristics for which neurons are split, the implications of splitting highly active versus occasionally active neurons etc.  I feel like the method has not been fleshed out enough for me to be confident that it is a general, robust and extensible framework.

There are also unanswered questions about how the choice of coding and non-coding neurons is made?  Is it 50-50?  Is there are more reasoned way of choosing this?  How are neurons allocated to c/nc during growth?  How is the network initialised in time, and how long is the network run for?

1.1.  Is it reasonable to expect the latent states of an RNN to reflect samples from a posterior?  I can't quite wrap my head around this, but I think that there should be some kind of decoder that maps the RNN state onto the samples?  Especially for neuro-plausibility, right?  I struggle to see why an SNN is the right architecture / amortised MCMC is the right test for this.  I invite the authors to comment.

1.2.  I also think ~80% of neurons are excitatory.  It would be nice if the SNN reflected that.

1.3.  I'm not sure how this would work, but it would be nice if there was a method for adding neurons in a better way.  I.e. if the network activity is low, add excitatory, if it is high, add inhibitory.  If a particular neuron is too active, add an inhibitory connection etc.

1.4.  Is the weight matrix dense?  Again, it would be nice if connections were sparse with some target sparsity / L1 regularisation.


### 2.  Presentation
A major point I am struggling with is the presentation of the method.  I believe the method itself is the idea of starting small and growing the network by adding new neurons.  I don’t fully understand why the presentation is so tied up in the GSM – which I think is just an example of where this could be applied, or indeed, a method for generating targets for supervised learning.  Sampling-based inference is also just an example application.  I feel like the intermingling of application and model/method in the exposition is maybe indicative of them being mingled in the authors minds?  It makes it very difficult for a reader to tease them apart, or indeed, how the authors intended them to be teased apart.  I wish the method was clearly presented in terms of inputs and outputs -- irrespective of how they were generated, their spaces, types, interpretation etc.

2.1.  Similar to this, but also related to (1) and (3), can you explain how the network actually performs inference, and how this inference is evaluated?  I understand it as the visual stimuli is repeatedly presented to the learned network as in (7).  Each timestep is then treated as a sample of the mean and covariance?  How do you then train this against the single and static values of $\mu_{GSM}$ and $\Sigma_{GSM}$?  How is this inference shown on Figure 2 & 3?  What are the different dots?  How are there different “Target” values?  What is the difference between red, blue and purple?  (And can we use more visually distinct colours please!)  Basically, F2, F3, the experimental task and the results need much more thorough explanation.

2.2.  I would also like to see an algorithm block or block-diagram of the method?  I think it is an appealing simple method, but a visualisation of this would still be nice.

2.3.  I also think calling it a “toolbox” is misleading – is at most a “method”, and, at least, a “heuristic”.

2.4.  If I understand correctly, the GSM is trained ahead of time to generate the supervised targets for learning the SSN?


### 3.  Empirical Evaluation
I find the empirical evaluation difficult to judge.  I am not too familiar with what a “standard” task is in this domain, but the task itself seems fairly abstract.  The performance metrics are also a little obscure and don’t have much explanation or justification.

3.1.  As suggested in (1), I’d like to see some ablations/variants of the method.  I would also like to see how repeatable the results are across random seeds.  I would like to see some form of quantitative error metric with error bounds to at least indicate how reliable this method is.  How many runs went unstable for each method during training?

3.2.  I also feel like the range of experimental evaluation is quite limited.  I understand that the network operates slightly differently to, for instance, AlexNet.  That said, i would still like to see how this network/method applied to slightly more “standard” NN tasks.  You certainly don’t need to achieve state of the art, but, if all i can reasonably use this for is MCMC in a particular model, then the scope and impact of the method is limited (which is a shame!).  This should be a method for generating a neural encoding for performing a task (be it classification off of the final value, average value in time, the raw samples etc).  It would also be interesting to see if this network can learn an encoding for time-series tasks.

3.3.  Unless I have misunderstood something, I feel like saying it is performing amortised MCMC is also a bit of a stretch.  It is essentially doing moment-matching using supervised learning.


# Minor Comments.

a.  Figure S2 is a cool figure, and I think would add real, concise impetus to the motivation of your method if included in the main.

b.  Why is the training loss so much higher for the larger network?

c.  Why do you do moment matching in (11)?  I would have thought you’d have done maximum likelihood, given that you can get out the distributions under the GSM.

d.  How are the Gabor filters computed and applied?  From Figure 1 it looks like there are multiple Gabor filters that are used, but there appears to only be one $A$ matrix.  It isn;t even really clear to me how (2) is used.  Where is $\phi$ used?

e.  I think the introduction can be slashed in length.  There is a lot of information in there that isn’t really relevant – get to the nub of the contribution:  a method for training neuro-plausible SSNs.  I also think Section 6 can be cut to make space.

f.  I don’t understand the “old stimulus” and “new stimulus” rows of Figure 3.  Can you explain them please?


# Very Minor / Typographical Comments.
i. Section headers should be in capital case, as opposed to sentence case.

ii. The abbreviation of excitatory and inhibitory to E and I makes those sentences harder to read.  I wouldn’t use the abbreviation.

iii. Putting the variable at the end of the square brackets for an expectation is non-standard and makes it harder to read.  They should be placed after the expectation, e.g. $\mathbb{E}_{z} \left[ \cdot \right]$.

iv. It would be handy if you linked the equations together more.  For instance, explicitly reference the pseudo-inverse in (12) to when it was introduced in (8).  It just helps tie the relevant math together.

v.  Line 98:  “optimise” -> “optimize”  (you have used US English elsewhere).

vi.  What is the subscript $F$ in the norms?  Frobenius norm?

vii.  In (12), how do you define the variance of a vector?  You take the variance within the vector?  Or the variance of elements between vectors?  Basically, $\mathbb{V}$ is undefined.

viii.  In (11), $\mathbb{C}$ is normally reserved for the space of complex numbers, as opposed to covariance.  Maybe use $\mathbb{C}\mathrm{ov}$ to be explicit.


# Summary.
Although my review comes off as quite negative, I do generally like the paper – I just do not think it is ready for publication at this point in time.  That said, if the authors can address my comments, then I would consider upgrading my review.

I implore the authors to explore theoretically justifying aspects of the method, or, performing further ablations/variations/experimental types to explore the method.  With these enhancements, this could be a very strong conference submission.

Good work, and good luck.

---

> ### Author Response · Authors · 2022-08-02
> **Response to reviewer 2UAi**
>
> Thank you for the helpful and constructive comments.
>
> 1.
>
> > the method is heuristic
>
> We also prefer theoretically principled over heuristic methods. However, we note that several current techniques in deep learning were introduced heuristically and became highly influential before (if at all) a deep theoretical principle has been found for them (eg. dropout [38k citations], or the progressive growing of GANs [5k]). Thus, we believe the NeurIPS community appreciates useful techniques, even if they are heuristic. Of course, further work is required to understand such techniques in rigorous theoretical terms.
>
> More importantly, the growth phase has always been theoretically-motivated. To better emphasize this, we have included a dedicated explanation in section 2.2.
>
> > exploration of "hyper parameters"/choices
>
>
> See 1.3 below.
>
> > network initialisation
>
> The network starts at resting membrane potential (which we define to be 0mV). The network is run for 500ms at timesteps of 0.2ms (for a total of 2500 time steps). These details have been added to the supplementary.
>
> 1.1 We agree that, in general, a more sophisticated decoder than our “identity” decoder (ie a one-to-one correspondence between coding neurons and latent variables) may be possible – indeed this has been explored eg. by Savin & Deneve (NeurIPS 2014). However, previous work found remarkably good predictions of neural responses by using a simple identity decoder of samples, including when samples were generated by an SSN just like in our work (Refs 35,38).
>
> 1.2 The Reviewer is correct in that there are more excitatory than inhibitory neurons in cortex, and our MNIST SSN model reflects this with a 4:1 excitatory:inhibitory ratio. The SSN trained for sampling-based inference remains at 1:1 for direct comparison with past approaches that have used that ratio.
>
> 1.3 We have now trained SSNs for two different tasks (MNIST and sampling-based inference) and have included some key reliability metrics that show the consistency of our method. For example, we built 100 separate networks for both tasks and found that none of the networks went unstable. We have also mentioned hyperparameter exploration as future work in the Discussion, as our main focus of keeping the network stable has been decisively achieved with a single set of hyperparameters.
>
> 1.4 Yes, see Discussion.
>
> 2.
>
> > presentation of the method
>
> As noted above (general response), we fully agree with this point. We have restructured the paper accordingly.
>
> 2.1 A constant visual stimulus is presented to the trained network. The membrane potentials of the neurons in the network then evolve over time. Moments (mean and covariance) are computed in each time step across trials. For training, we compute the error of each of these moments against the single static target moments, and average errors across time steps. We have since obtained similar results using the converse approach, by computing moments across time (within trial) and averaging errors across trials (not shown). See also new section 4.3 and a dedicated section in the supplementary (see Appendix B).
>
> 2.2. See new Fig. 1.
>
> 2.3. We agree and will change the title (subject to AC approval). We also refer to it as our “method” in the main text.
>
> 2.4. That is indeed the case.
>
> 3.
>
> > empirical evaluation
>
> We have included additional detail in section 4.3.
>
> 3.1. We have introduced three additional reliability metrics for this. E.g. one  metric shows that no runs went unstable during training for either task.
>
> 3.2 Thanks for the suggestion. We have now trained an SSN to perform the classic MNIST classification task, showing that our method for training SSNs generalizes for different tasks.
>
> 3.3. Our trained SSN is capable of generalizing to new targets, unseen during training, so we believe it qualifies as performing amortized approximate MCMC in line with previous such approaches (eg. Ref. 35).
>
> Minor Comments
>
> a. We have introduced the quantities on those plots as “reliability metrics” in the main text.
>
> b. The larger network is trained on a more complex GSM, which is harder to fit. We now consolidated all model details in section 4.3, and a dedicated section in the supplementary (see Appendix B).
>
> c. For direct comparison with previous approaches (Ref. 35).
>
> d. We have updated equation (9) to show where $\phi$ is, and explained how matrix A includes multiple Gabor filters in section 4.1 of the main text.
>
> e. We have slightly reduced the length of the introduction and have incorporated the limitations in (what was previously known as) section 6 in the discussion section.
>
> f. We have now removed this figure in the revised submission as it is irrelevant to the new focus of training SSNs.
>
> Very minor comments have all been addressed. We thank the reviewer for pointing them out and for all the constructive comments above.

---

> > ### Comment · Reviewer_2UAi · 2022-08-07
> > **Rebuttal Response**
> >
> > To the authors,
> >
> > Sorry for the delay in my response.  Thank you for replying to my review in such detail.
> >
> > The authors are to commended for restructuring the entire paper and conducting a raft of new experiments, as per the requests of the reviewers.  I think the new version is unquestionably stronger in just about every respect.
> >
> > I do stand by my comment that it is still a heuristic.  The authors have convinced me that this is not a terminal flaw, but I equally do not this area a "strength" as a result.
> >
> > I also stand by the comment (also raised by another reviewer) that the introduction is _far_ too long.  I would also like some of the MCMC figures to be brought back in to the main text, but maybe simplified and enlarged.  For any potential accepted version, the extra page can be used for this (and for explaining the experiment in more detail).
> >
> > This wasn't in my original review, but it might also be nice to include figures showing the time-evolution of the neural state during execution (as opposed to the scatter plots originally included).
> >
> > Fwiw, I also support the authors being permitted to change the title.  I think the version the authors have changed it to (_"Deep Learning in Stochastic Stabilized Supralinear Networks"_) is too general.  I would prefer something along the lines of _"Improved Training of Stochastic Stabilized Supralinear Networks"_ (although I concede that is a little vanilla).  Your main contribution is in the training methodology.
> >
> > I also echo pretty much all of the comments Reviewer rMNf made in their response [https://openreview.net/forum?id=znbTxnBPlx&noteId=OMYTOmMsc7](here), particularly on understanding the new MNIST experiment in light of other SSN / MLP methods.
> >
> > Given that the authors resolved a good number of my concerns, but that there are some outstanding concerns, I will upgrade my review from a three to a five.
> >
> > Thank you very much, and good luck,
> > 2UAi

---

> > > ### Author Response · Authors · 2022-08-07
> > > **Closing comments for Reviewer 2UAi**
> > >
> > > We thank the Reviewer for the encouraging review and improved score. We have already updated our paper on 6th August in response to the comments echoed from Reviewer rMNf which includes a shortened introduction, discussion and conclusion, and with all experimental issues addressed. We will include more information about the dynamics of the SSN and how it performs MCMC in the additional page if permitted. We agree with the opinions on the title and will give further thought, as similarly permitted.
> > >
> > > Thank you

---

> > > > ### Comment · Reviewer_2UAi · 2022-08-08
> > > > **Clarification**
> > > >
> > > > Thank you to the authors.  Just to clarify, I still think there is some work to be done in further understanding the MNIST example - I would like to see different networks of different sizes trained and compared to a suitable benchmark SSN method and an MLP method (ie what is the performance deficit and exactly where does your method outpace other methods in terms of network size), I would like to see what the latent dynamics are, how many neurons are used and how many are “dead” etc.  Basically, I would like to see the dynamics of network training and the trained networks themselves explored a little bit more.  Especially to bring you towards the originally outlined neuroscience goals.  I also appreciate you restructured the paper a lot, but I think the intro and discussion are still too long, but maybe that is personal preference.
> > > >
> > > > Either way, I will maintain my score from here.  Good luck.

---

> > > > > ### Author Response · Authors · 2022-08-09
> > > > > **Thank you**
> > > > >
> > > > > Thank you for the clarification. We will make the best out of the additional page if permitted, and prioritize including more information about training over the introductory section as required.

---

### Author Response · Authors · 2022-08-02
**General response to reviewers**

We thank all three Reviewers for their replies. All Reviewers have independently expressed a similar sentiment that the wider NeurIPS community would better appreciate the paper if its main focus was on the training of SSNs, and that we should decouple the niche topic of sampling-based inference under the GSM from the introduction and main narrative. We fully agree with this point and have completely restructured our paper. We briefly highlight the main changes to the paper here in addition to our specific Reviewer responses:

1. The main narrative is now rightfully focused on our method of training SSNs, and less relevant details about the GSM model setup have been moved to the supplementary.

2. We have trained the SSN to perform an additional task of MNIST classification to address the concerns about whether SSNs can be generalized to perform other tasks.

3. We have included three reliability metrics in our analysis to show how well our method solves the problem of instability.

4. We have consolidated information regarding past approaches of training the SSNs to section 4.3 and a dedicated section (Appendix B) in the supplementary to address any confusion about past and current approaches.

5. We hope that with these changes the paper will meet with the approval of the Reviewers and the Area Chairs.

Edit (3rd August): Fixed typos in supplementary.

---

> ### Author Response · Authors · 2022-08-06
> **Edit (6th August 2022)**
>
> We have made further changes after the latest comments from reviewer rMNf:
> - In all our experiments, we rightfully use standard train/val splits with all our standard datasets (MNIST and CIFAR-10)
> - We have added much clearer comparisons that will help with seeing the effectiveness of our method, as well as the scale of improvement our method provides over past approaches:
>
> MNIST: We optimize both our model and past models to directly compare the approaches, and also have an MLP and logistic regression as base comparisons.
>
> GSM Inference: We optimize both our model and past models on both our GSM and past GSM models. (So all 4 combinations of old-old, old-new, new-old, new-new)
>
> We hope this these results will convince the reviewers.

---

### Meta-Review · Area_Chair_ECVD · 2022-08-30

**Recommendation:** Accept
**Confidence:** Less certain

**Metareview:**

This paper proposes a method for training stochastic stabilized supralinear networks (stochastic SSNs), a theoretical model that has shed light on how circuits of excitatory and inhibitory neurons perform various computations. Compared to RNNs commonly used in machine learning, SSNs are trickier to train since they have unbounded nonlinearities and no gating mechanisms. The authors devise a training scheme that adds neurons one at a time while preserving stability, and they demonstrate how their method can learn networks that perform inference in a commonly-studied Gaussian scale mixture model.

The reviewers improved their scores over the course of the discussion, and with the authors’ revisions the paper should be accepted. The main criticisms were that the audience is somewhat narrow and the method is a bit heuristic. Neither is a "terminal flaw," to use one reviewer's words, but I would encourage the authors to consider ways to improve on these aspects for the final version.

**Award:**

No

---

### Decision · Program_Chairs · 2022-09-14

Accept